

# Assessment of nationally representative dietary studies in the Gulf Cooperation Council: a scoping review

Rukshana Hoque[1], Erin Strotheide[2], Juliann Saquib[3] and Nazmus Saquib[3]

[1] Diabetes & Nutritional Sciences Division, King's College London, London, United Kingdom
[2] Research Unit, Sulaiman Al Rajhi University, Bukayriah, Al-Qassim, Saudi Arabia
[3] College of Medicine, Sulaiman Al Rajhi University, Bukayriah, Al-Qassim, Saudi Arabia

## ABSTRACT

**Background.** Obesity is at a record high in Gulf Cooperation Council (GCC) countries and is expected to continue increasing. Diet is a major contributor to this disease, but there is inadequate nationally representative dietary research from these countries. The aim was to quantify the number dietary studies using food frequency questionnaires (FFQs) that have been conducted in individual GCC countries and to assess the quality of eligible studies.

**Methodology.** Four databases (PubMed, Web of Science, MEDLINE, and DOAJ) were searched for keywords; records were screened for eligible studies and data were abstracted on study characteristics (publication year, geographical locations, sample size, units of measurement, number of foods examined, number of Arab foods and key findings). Quality was assessed using an adapted Newcastle-Ottawa Quality Assessment Scale for cross-sectional studies.

**Results.** Only seven studies were eligible from four of six GCC countries (Saudi Arabia, Bahrain, Kuwait and Qatar). All eligible studies used FFQs, but only 29% used a validated questionnaire, one being in Arabic, and none of the studies used any additional tools to measure diet. Fifty-seven percent of studies made an effort to include local foods. The majority of studies (71%) either measured frequency or quantity of food consumed, but only 29% attempted to account for both frequency and quantity.

**Conclusions.** The quality of studies varied and major weaknesses of FFQ validity and adaptability have been highlighted. More dietary investigations are needed using validated FFQs that have been adapted to the local GCC diets. Using reference tools will allow for better dietary estimations.

Corresponding author
Nazmus Saquib, a.saquib@sr.edu.sa

# INTRODUCTION

Obesity is an epidemic in the countries of the Gulf Cooperation Council (GCC) (that is, Saudi Arabia, Bahrain, Kuwait, Oman, Qatar, and United Arab Emirates). Approximately one out of every three adults is obese (Body Mass Index ≥30), and the obesity prevalence has been rising in every member country. For example, between 2011 and 2016, the obesity prevalence rose in Saudi Arabia (KSA) from 32.1 to 35.4%, in Bahrain from 27.1% to

29.8%, in Kuwait from 35.1% to 37.9%, in Oman from 23.7% to 27%, in Qatar from 31.8% to 35.1%, and in the United Arab Emirates (UAE) from 28.3% to 31.7% (*Global health observatory, 2017*). Apart from obesity, the GCC countries are also leading countries in the world in diabetes and cardiovascular disease prevalence (*M Alqarni, 2016*; *Alqurashi, Aljabri & Bokhari, 2011*; *Aljefree & Ahmed, 2015*).

There is mounting evidence of a potential causal link between specific dietary factors (such as, fruit, vegetable, processed meat, and trans-fat intake) and the above mentioned chronic conditions (*Micha et al., 2017*; *World Cancer Research Fund, 2018*; *World Cancer Research Fund, 2010*). A recent systematic review of dietary data from 195 countries found that 22% of all adult deaths worldwide are due to unhealthy diet; more than half of diet-related deaths are attributable to a high sodium intake, low intake of whole grains, and low fruit intake (*GBD 2017 Diet Collaborators, 2019*).

Several factors likely contribute to obesity in GCC countries. With increased wealth from oil reserves, these countries have seen rapid economic growth. The urbanization of the landscape has seen a rise in international fast food chains, making it easier and quicker to consume processed foods (*Alnohair, 2014*; *Al-Mahroos & Al-Roomi, 1999*). This has resulted in a change of diet from traditional, locally produced goods such as wheat, vegetables and dates to fast foods high in fat, sugar and salt content (*Al-Othaimeen, Al-Nozha & Osman, 2007*). Whist all GCC countries have attempted to develop a national plan that addresses nutrition and physical activity, most have not followed up, which makes it difficult to evaluate the impact of such programs (*Samara, Andersen & Aro, 2019*). Changes in lifestyle such as increased use of cars, electrical home appliances, television and gaming devices have resulted in a more sedentary lifestyle (*Alnohair, 2014*; *Musaiger, 2004*). The extremely hot climate found in these countries also likely deters outdoor activities with people opting to use cars, even for short journeys (*Alnohair, 2014*; *Al-Kandari, 2006*). A combination of all these is likely to play a role in the current epidemic.

Given the high prevalence of chronic conditions in the GCC, one would expect that these countries engage extensively in diet and nutrition research. However, dietary studies have been limited; only approximately 1% of global dietary research has come from Arab countries (*Sweileh et al., 2014*). Their *h-indices* [measurement of performance by combining productivity (number of papers) and impact (number of citations)] are much lower than neighbouring non-Arab countries (*Hirsch, 2005*).

One would similarly expect that assessment tools used in dietary studies from GCC countries would differ from those in European or North American studies as Middle Eastern diets vary a great deal from their Western counterparts. For example, date palm fruit is highly consumed in Gulf regions, with daily consumption ranging from 68–164 g daily (*Al-Mssalle, 2018*; *Ismail et al., 2006*; *Aleid, Al-Khayri & Al-Bahrany, 2015*), whereas only 140 g of this fruit is consumed *annually* in Europe (*Ordines, 2000*). Differences such as these should be accommodated for when designing dietary assessment tools.

The usual assessment tools used in dietary research are 24-hour dietary recall (open-ended, food consumed the previous day, conducted by trained interviewer), diet records (open-ended, participants trained to record own diet), and food frequency questionnaires (FFQs) (closed-ended, typically a food list and frequency of consumption in a given

period). All have strength and limitations (*Shim, Oh & Kim, 2014*), but due to low cost, low respondent burden and ease of use compared to other methods, FFQs are thought to be the best choice for measuring habitual diet in large populations. The usefulness and reliability of FFQs have been demonstrated with strong correlations with diet records (*Rimm et al., 1992*; *Willett et al., 1985*), dietary recalls (*Katsouyanni, 1997*; *Brunner et al., 2001*; *Marques-Vidal et al., 2011*), and objective biomarkers of diet (*Katsouyanni, 1997*; *Brunner et al., 2001*). As an FFQ is a self-reported subjective tool, FFQs should be tested for validity alongside a reference tool.

The authors' aimed to conduct both a quantitative and qualitative review of all dietary studies conducted within each GCC country. To be as nationally representative as possible and to provide a current and more reflective picture of diet in the GCC, only studies carried out in multiple regions (must be a minimum of two regions) were included. Dietary research that used FFQs in individual GCC countries (Bahrain, Kuwait, Oman, Qatar, KSA, UAE) over the past ten years (2009–2019) were assessed. The characteristics of the studies were described and their quality was assessed using a widely accepted scoring tool (*Herzog et al., 2013*; *Stang, 2010*). The objectives were to (1) identify multi-regional GCC dietary studies that used FFQs, (2) assess the quality of the studies, and (3) offer recommendations for future dietary assessments.

## METHOD

### Search strategy and inclusion criteria

This review was conducted in May 2019. PubMed, Web of Science, MEDLINE, and Directory of Open Access Journals (DOAJ) databases were searched using the following terms: "diet," "frequency questionnaire" in combination with each of the Gulf Cooperation Council countries ("Bahrain", "Kuwait", "Oman", "Qatar", "Saudi Arabia", "UAE"). A total of 431 records were identified from PubMed ($n = 241$), Web of Science ($n = 34$), MEDLINE ($n = 132$) and DOAJ ($n = 24$). Duplicates ($n = 39$) were removed, and the unique records ($n = 392$) were screened for the following inclusion criteria: (1) assessed diet using a food frequency questionnaire, (2) included data from multiple regions/cities (minimum two) of the Gulf country of focus, and (3) data were collected in the last ten years (that is, 2009 and later).

### Exclusion of studies

Studies were excluded if they (1) examined data from only one specific region/city/population group and therefore were not necessarily nationally representative, (2) were multinational studies that did not give Gulf-nation-specific results, (3) were not conducted in a GCC country, (4) were intervention studies where the diet had purposefully been changed, (5) were review or meta-analysis papers, (6) used an assessment tool other than a food frequency questionnaire, or (7) had no findings related to diet or did not report those findings. Therefore, the final analysis was limited to seven dietary studies (Fig. 1).
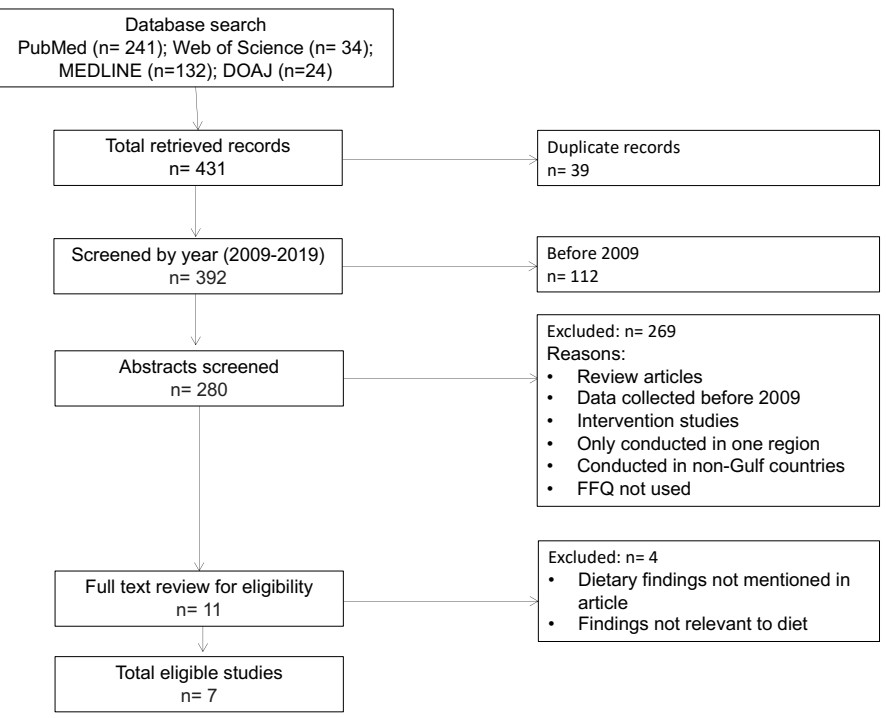

**Figure 1   Flow chart of study eligibility of dietary studies conducted in GCC countries.**

## Data charting process

After an initial search and screening, the following data from each study were charted:
publication year, author(s) name(s), geographical location, sample size, age range
of participants, dietary assessment tool(s) used, units of measurement (for example,
times/week, servings/day, etc.), total number of foods examined, number of Arab-specific
foods (and where possible, the type and name of food), whether the questionnaire was
validated, and dietary findings related to the most common foods studied. Any discrepancy
was resolved through discussion and consensus among the authors.

## Critical appraisal of studies

Using a scoring system adapted from the Newcastle-Ottawa Quality Assessment Scale for
cross-sectional studies (*Herzog et al., 2013*), each study was scored for (1) representativeness
of the sample, (2) sample size, (3) non-respondents, (4) ascertainment of the exposure, (5)
adaptability, (6) assessment of the outcome, and (7) statistical test (Appendix S1).

## Data analysis

Study characteristics, along with main findings related to dietary intake/habits were
tabulated. Additionally, indicators of study quality were assigned point values based on the
quality assessment scoring scale and then summed. Each study was categorized as excellent
(9–12 points), satisfactory (5–8 points), or unsatisfactory (0–4 points).

## RESULTS

### Study characteristics

The search resulted in seven studies published between 2009 and 2019. Tables 1 and 2 show three studies were conducted in Saudi Arabia, one in Kuwait, one in Bahrain, and two in Qatar; there were no studies from Oman or the UAE. A majority of the studies ($n = 6$) had sample sizes greater than 1,000 participants, and all studies included a sample size justification. Almost all studies had a 1:1 male: female ratio (range 1: 0.9–1.4 male: female). Fifty-seven percent ($n = 4$) of the studies were carried out with adolescents (12–19 years of age), whereas 33% ($n = 2$) included both adolescents and adults. One study (14%) classified participants 18 years and older as adults, thus the study was considered to be carried out on an all-adult population (*Donnelly, Fung & Al-Thani, 2018*).

All studies used FFQs, but three administered the FFQ through face-to-face interviews; the rest were self-administered. One study (*Moradi-Lakeh et al., 2017*) used pictures to deduce serving sizes.

The number of food items assessed ranged from two (non-specified fruits and vegetables) (*Haj Bakri & Al-Thani, 2012*) to twenty items (*Donnelly, Fung & Al-Thani, 2018*). Only two of seven studies used a validated questionnaire, adapted it for local cuisine, and had it pilot tested for suitability (*Donnelly, Fung & Al-Thani, 2018*; *Musaiger et al., 2011*) and from these, only one was conducted in Arabic; the other five studies did not use validated FFQs.

Key findings from each study varied based on the units of measurement. Frequency ranged from days per week, times per day, servings per day, to categories (such as, always, sometimes, never). Quantity options were servings per day, serving sizes, and serving sizes via selection of pictures.

### Quality assessment of studies

Only one of all included studies used a validated Arabic questionnaire (all were presented in English in the article) and none used any additional tools to measure diet. *Donnelly, Fung & Al-Thani (2018)* translated the questionnaire to Arabic and back to English to ensure correct language usage. For relevance to local contexts, focus groups were conducted and the questionnaire pilot tested and thereafter further refined. *Musaiger et al. (2011)* modified a previously validated questionnaire (Family Eating and Activity Habits Questionnaire) (*Golan & Weizman, 1998*) and adapted it to ensure it reflected dietary habits of the target population. Contents of the FFQ were validated by experts in the field of nutrition, public health, and epidemiology and the questionnaire underwent pilot and test-retesting (*Musaiger et al., 2011*).

Table 3 shows 57% ($n = 4$) of the studies made an effort to include local foods, scoring a point for adaptability, whereas the other three studies either did not incorporate any local foods or did not mention it in their studies.

Five studies measured either frequency or quantity, whilst two studies scored the maximum three points for 'assessment of outcome' by having units of measurement that took into account both frequency and quantity (that is, times/week *and* servings/day).

All studies used appropriate statistical analysis and 86% ($n = 6$) had an adequate response rate ($\geq 60$%). One study had a 52.1% response rate (29) and one study did not

Hoque et al. (2020), *PeerJ*, DOI 10.7717/peerj.10163

**Table 1** **Background information and characteristics of Gulf Cooperation Countries (GCC).**

| | Bahrain | Kuwait | Oman | Qatar | Saudi Arabia | UAE |
|---|---|---|---|---|---|---|
| Year country was founded/ independent | 1971 | 1961 | 1951 | 1971 | 1932 | 1971 |
| Surface area ($km^2$) | 774 | 17,188 | 309,500 | 11,628 | 2,149,690 | 77,700 |
| Population (thousands) in 2016 | 1425 | 4053 | 4425 | 2570 | 32,276 | 9250 |
| Obesity prevalence in 2011 (%) | 27.1 | 35.1 | 23.7 | 31.8 | 32.1 | 28.3 |
| Obesity prevalence in 2016 (%) | 29.8 | 37.9 | 27 | 35.1 | 35.4 | 31.7 |
| Net change in obesity (%) | +2.7 | +2.8 | +3.3 | +3.3 | +3.3 | +3.4 |
| Total number of hits using keywords[a] | 29 | 64 | 56 | 52 | 176 | 46 |
| Studies included | **1** Musaiger et al. (2011) | **1** Al Baho & Badr (2011) | **0** | **2** Haj Bakri & Al-Thani (2012) Donnelly, Fung & Al-Thani (2018) | **3** Moradi-Lakeh et al. (2017) Al-Hazzaa et al. (2011) AlBuhairan et al. (2015) | **0** |

**Notes.**
[a]PubMed; Web of Science; MEDLINE; DOAJ (Directory of Open Access Journals).

Hoque et al. (2020), *PeerJ*, DOI 10.7717/peerj.10163

**Table 2  Study characteristics of national dietary assessment studies conducted in Gulf Cooperation Countries (GCC) ($n = 7$).**

| Author | Country | Age range | Sample size | Tool(s) used | | | | | Findings |
|---|---|---|---|---|---|---|---|---|---|
| | | | | Type | # of total food items | # and type of Arab food[a] | Measurement | Validated | |
| Al Baho & Badr (2011) | Kuwait | 13–15 | 2674 (1399 male; 1275 female) | FFQ (2011 Kuwait GSHS) | 6 (includes breakfast meal) | 2 *Coriander* (vegetable); *KDD, KDcow, Carnation* (dairy) | times/day in past 30 days (except breakfast: how often in last 30 days: Never, Rarely, Sometimes, Mostly, Always) | Not validated | Over 30 days, 36% of students usually ate fruits ($\geq 2$ times/day); 19% ate vegetables ($\geq 3$ times/day); 75% consumed soft drink ($\geq 1$ times/day); 36% drank milk ($\leq 2$ times /day); 48% had fast food ($\geq 3$ times/week). |
| AlBuhairan et al. (2015) | Saudi Arabia | 12–19 | 12575 (6444 male; 6131 female) | FFQ (Global School-based Student Health Survey) | 8 (includes meals) | 2 *Fatayer* (snack); *molokhiya* (vegetable) | srvgs/day breakfast: last 30 days (never, rarely, some, most, daily) Number of main meals per day? (0 - >4) | Not validated | 38% of adolescents ate $\geq 1$ srvgs/day of fruit and 54.3% ate $\geq 1$ srvgs/day of vegetables. 38% drank $\geq 2$ carbonated beverages/day. |
| Al-Hazzaa et al. (2011) | Saudi Arabia | 14 - 19 | 2908 (1401 male; 1507 female) | FFQ (Arab Teen Lifestyle Survey (ATLS)) | 9 (includes meals) | None | days/wk | Not validated for dietary questions | In Saudi adolescents, an average of 22.8% consumed vegetables daily; 12.8% had fruit daily; 29.15% had milk daily; 62.35% consumed sugar-sweetened beverages (SSB) (>3 day/week); 27.55% fast food (>3 day/week); 27.85% french fries/potato chips (>3 day/week); 26.8% cake/donut/biscuit intake (>3 day/week); 44.95% sweets/chocolates intake (>3 day/week); 50.65% energy drinks intake (>3 day/week). |
| Donnelly, Fung & Al-Thani (2018) | Qatar | $\geq 18$ | 1606 (804 male; 802 female) | FFQ | 20 | 1 *Shawarma* (meat products) | Never, Seldom, times/wk, once or more daily | Validated | Participants ate fruits (35.8%), green vegetables (31.8%) and other vegetables (44.1%) at least once daily. 44.7% consumed milk products and 14.4% drank carbonated soda more than once daily. 26.1% of participants on average ate pasta, snacks and cakes or pastries 2–4 times/week. An average of 32% consumed protein products 2–4 times/week. |

Hoque et al. (2020), *PeerJ*, DOI 10.7717/peerj.10163

**Table 2** (*continued*)

| Author | Country | Age range | Sample size | Tool(s) used | | | | | Findings |
|---|---|---|---|---|---|---|---|---|---|
| | | | | Type | # of total food items | # and type of Arab food[a] | Measurement | Validated | |
| Haj Bakri & Al-Thani, 2012(31) | Qatar | 18–64 | 2496 (1053 male; 1443 female) | FFQ via face-to-face interviews STEPS Instrument (WHO 2005) adapted for Qatar-specific context) | 2 | None | days/wk AND srvgs/day | Not validated | 91% of the Qatari studied population consumes <5 srvgs/day of fruits and/or vegetables. Average number of fruit servings was 0.8 srvgs/day. Average number of vegetable servings was 1.4 srvgs/day. Overall average combined fruit and/or vegetable servings was 2.2 srvgs/day. |
| Moradi-Lakeh et al. (2017) | Saudi Arabia | 15–60+ | 10735 (5253 male; 5482 female) | FFQ via interview; pictures of serving sizes | 14 | 2 *Laban* and *labneh* (yogurt products) | days/wk in the last year AND g/day or ml/day | Not validated | 11% of subjects ate fruits daily and 26% ate vegetables daily. 27% drank SSB daily. Dietary guideline recommendations for fruits were met by only 5.2% of participants and 7.5% for vegetables. 85% met the recommended intake for meat and 80% met recommendations for processed meats. |
| Musaiger et al. (2011) | Bahrain | 15–18 | 735 subjects (339 male; 396 female) | FFQ | 18 (includes meals) | None | times/wk  fast food/soft drinks: times/wk AND typical srvg size meals: regularly (yes/no) snacking: always, sometimes, never | Modified from validated questionnaire and pilot-tested | Approximately 25% of respondents reported eating fruit daily, 27.7% consumed fruit rarely (<1 time/week). 26% consumed vegetables daily, 38% of respondents rarely (<1 time/week). 37% consumed dairy products daily; 22% rarely (<1 time/week). 20% consume meat daily; 21.5% rarely (<1 time/week). 14.4% of participants ate fast food daily, 29% rarely (<1 time/week). Soft drinks: 42.2% of participants consume soft drinks daily; 27.8% rarely (<1 time/week). |

**Notes.**

[a]where possible, names of Arab food have been included.

[#]number,

SSB, sugar sweetened beverages.
**Table 3  Quality assessment of national dietary assessment studies conducted in GCC countries using a scoring system ($n = 7$).**

| Author | Design | Representative of sample | Sample size | Selection Non-respondents | Ascertainment of exposure (validated) | Adaptability | Outcome Assessment of outcome | Statistical test | Total Score (out of 12) |
|---|---|---|---|---|---|---|---|---|---|
| Al Baho & Badr (2011) | cross-sectional | + | + | + | + | + | + + | + | 8 |
| AlBuhairan et al. (2015) | cross-sectional | + | + | + | + | + | + + | + | 8 |
| Al-Hazzaa et al. (2011) | cross-sectional | + | + | | + | | + + | + | 6 |
| Donnelly, Fung & Al-Thani (2018) | cross-sectional | + | + | | +++ | + | ++ | + | 9 |
| Haj Bakri & Al-Thani (2012) | cross-sectional | + | + | + | + | | + + + | + | 8 |
| Moradi-Lakeh et al. (2017) | cross-sectional | + | + | + | + | + | + + + | + | 9 |
| Musaiger et al. (2011) | cross-sectional | + | + | + | + + | | + + + | + | 9 |

compare between respondent and non-respondent characteristics or take non-responses into account (or did not mention it in their study) (*Al-Hazzaa et al., 2011*).

## DISCUSSION

With such a high prevalence of diseases to which diet is a major contributor, it is surprising that there are so few multi-regional studies that investigated diet in the GCC in the past ten years. Five out of the seven studies included in this review did not use validated FFQs.

Dietary summaries show intake of fruit and vegetables being far below the recommended three servings of vegetables and two servings of fruit per day (*Ministry of Health Saudi Arabia, 2012*). In Saudi Arabia, only 5.2% of individuals met the recommendation for fruit intake and 7.5% for vegetable intake. In contrast, consumption of sugary beverages was oversubscribed, with an average of 36% of adolescents (14–19 years old) reporting daily consumption (*Al-Hazzaa et al., 2011*) and 27% of 15–60 year olds (*Moradi-Lakeh et al., 2017*), exceeding local and global recommendations of sugary-drink consumption (*Eaton et al., 2012*; *Al Qaseer, Batarseh & Asa'ad, 2007*; *AlBuhairan et al., 2015*). This low fruit and vegetable intake, combined with high sugary-beverage consumption, suggests a poor-quality diet across the GCC.

The varying methods of measuring diet made it difficult to compare consumption. For example, 57% of the studies assessed diet using frequency questions (how often), whilst 43% measured frequency and quantity (portions or serving size). At times, the response categories were too broad for in-depth analysis. For example, "Do you regularly consume meals? Yes/No" (*Musaiger et al., 2011*) does not specify which meals, how many meals, or the content of the meals. Similarly, "How often do you drink a glass of milk?" (*Alsheridah & Akhtar, 2018*) does not quantify the size of the glass or the amount of milk consumed.

Adaptability was one of the main issues relating to study quality according to the quality assessment scoring scale. Studies need to make it explicit how they have categorized foods, for example, whether they have classified potatoes as starch, tuber, snack, fast food, etc. Four studies attempted to include local foods, with a maximum of two or three items added (and mentioned in the article) (*Donnelly, Fung & Al-Thani, 2018*; *Moradi-Lakeh et al., 2017*; *AlBuhairan et al., 2015*; *Al Baho & Badr, 2011*). It is concerning that the other three studies did not mention any native foods at all. In Tabacchi's review (*Tabacchi et al., 2014*), it is suggested that an FFQ with less than 70 food items reduces the quality of nutritional information that can be deduced. None of the studies included in this review had 70 items; the most was 20, the average being 11 items. Nutritional status and dietary patterns differ over time and from region to region; without the incorporation of local foods and without categorizing them under more common food groups, it is entirely possible to mask important epidemiological links between diet and disease.

An overall poor validity of FFQs was found in this review. Only one study used a validated Arabic FFQ and scored three points out of a possible four points on the quality assessment scale. Validation in large-scale studies is especially important as FFQs are prone to measurement errors and come with inherent self-bias. FFQs rely on an individual's memory and his/her own perception of food sizes, thus under-reporting remains a common

problem (*Subar et al., 2003*; *Beaton et al., 1979*; *Freudenheim & Marshall, 1988*; *Kipnis et al., 2003*). Researchers have made extensive efforts in the last two decades to mitigate some of the errors with self-reporting data (*Freedman et al., 2014*; *Freedman et al., 2010*; *Kipnis et al., 2002*), but diet and eating patterns are complex, and FFQs are still thought to have clear value and insight that solely objective measures cannot provide (*Kirkpatrick et al., 2017*; *Subar et al., 2015*). One of the ways to minimize errors is to use a validated FFQ. FFQs are not one-size-fits-all, and it is integral that questionnaires be adapted/modified to suit the population with which they are being used. This includes first developing a good FFQ to standard procedure (*Willett, 2012*), FFQs being in the native language, which for GCC is predominantly Arabic, and including as many local foods as possible.

Within obesity research, two areas are deficient: understanding the role of dietary habits in the obesity epidemic and sufficient intervention studies on weight loss via dietary change. Research on dietary habits in the obesity epidemic may be lacking due to a shortage of skilled researchers and research centers (*Samara, Andersen & Aro, 2019*). Obtaining accurate dietary data requires specialized nutritionists/dieticians and controlled research settings, but this is a problem across many Gulf states, where it is difficult to have sufficient numbers of well-trained staff to serve large populations and areas like Saudi Arabia (Table 1). Investments should be made in specialized university health education and research courses and training in hospital departments; this will take time and resources but is a necessary step to produce expert personnel that can adequately face the challenges of regional obesity research (*Samara, Andersen & Aro, 2019*).

Research in this field may also be looking at risk factors found in Western countries and not necessarily exploring factors that are unique to the socio-cultural environment of the GCC. For example, women have been shown to be less active than men across Gulf countries (*Al-Nozha et al., 2007*; *Mabry et al., 2009*) and more sedentary than their British counterparts (*Al-Hazzaa et al., 2013*; *Al-Nakeeb et al., 2012*), but reasons for this behavior was poorly understood. Only by exploring the socioeconomic, environmental and cultural contexts further was it understood that the greatest barrier to physical activity for women was a lack of facilities rather than assumed low levels of knowledge, dress codes (*Samara et al., 2015*) or high obese-body acceptance (*Wills et al., 2006*). Samara et al. suggested that future health strategies should focus on providing culturally sensitive exercise facilities for women (*Samara et al., 2015*). A similar approach needs to be taken for nutrition and diet, where interventions, based on survey results, acknowledge and work with, not against, local culture and social norms (*O'Dea, 2008*). Such intervention studies need to have tangible goals, clear action plans and sufficient follow-up to evaluate long-term effectiveness (*Lawton et al., 2006*).

Limitations of this review are that the search was carried out on four main databases; this may have missed studies published in other journals not found within these databases, and those that are currently underway or not yet published. However, additional cross-checking was performed with reference lists to ensure the maximum number of studies were screened. The small number of studies limited the generalizability of findings. To the authors' knowledge, there are no other reviews similar to the current study. There are studies that have looked at other methods for country-specific dietary assessment

(*Kirkpatrick et al., 2017*) and the Newcastle-Ottawa Assessment Scale, which was adapted for this study, has been used to assess study quality but not in the same context as the current study (*Herzog et al., 2013*; *Modesti et al., 2016*). Finally, although studies have looked at dietary research from other parts of the world, no study has quantified the number of dietary studies coming specifically from the GCC and assessed their quality. Our review is unique in these ways, so the results of this present study cannot be easily compared to other studies.

A particular strength is the quality assessment aspect of this review. Adapting a scoring system allowed for objective assessment of studies. It highlighted that most of the included studies were either satisfactory ($n = 4$) or excellent ($n = 3$), whilst making it clear that the greatest weaknesses were in the number of food items and the validity and adaptability of FFQs, which researchers should take into consideration when designing future studies. Another strength is that the review focused on large-scale, multi-regional studies, which are more representative of the respective GCC nations' populations.

## Recommendations

As validity and adaptability were the lowest scoring categories, it is important to address this.

1. Validation can be assured by using a reference method. There are a variety of other methods used to measure diet, including self-reporting food records and 24-hour dietary recall (24-HDRs), but the most objective reference tool is food or nutrient biomarkers (*Shim, Oh & Kim, 2014*; *Hedrick et al., 2012*). In theory, biomarkers look like a promising method to remove the human error that comes with self-reported dietary data, but their widespread use is hindered because there are only a few known and validated biomarkers. One of the well-known biomarkers could be used as a reference measurement to validate FFQs and to assess their accuracy.

2. As KSA is the largest of the GCC countries, a quality assessment of all FFQs used in KSA should be undertaken. Comparisons should be made to see how similar they are, how inclusive they are of local cuisine and if the questionnaires are validated. This will be a labour-intensive task as the questionnaires are rarely attached to the articles or submitted as supplementary material; thus, authors will need to be contacted for original FFQs. This will give an overview of the versions of FFQs available and the Arabic food items included. By noting what foods are *not* represented in these questionnaires, additional foods can be added and attempts made to validate the FFQ. A recent FFQ developed by *Gosadi et al. (2017)* is a promising start for KSA. The Arabic FFQ had 140 food items and ensured it had a comprehensive food list by comparing it with open-ended information from 24-hour dietary recalls to find that 85% of food items recalled were covered in the FFQ. The FFQ has been piloted and its reliability assessed (Cronbach's alpha test and test-retest) and it should now be used in other regions. This standard of FFQ development should be carried out with other GCC countries as well to better capture dietary habits.

3. The review only included cross-sectional studies because they give a current picture of diet (observations of diet at a given point in time). Carrying out a longitudinal study analysis (repeated observations of a population over time) would illuminate how diet has changed over time to make better-informed future predictions.

## CONCLUSIONS

This is the first review to collect, quantify and critique the quality of data from dietary studies conducted in GCC countries by using an objective scoring system approach. Study quality varied, and major weaknesses of FFQ validity and adaptability have been highlighted.

Findings consistently showed that the majority of GCC populations are not meeting the recommended fruit and vegetable intake, and sugary-beverage consumption is on the rise, implying a poor diet. However, interpretations are made with caution due to the low study sample included ($n = 7$). In these GCC countries, where obesity levels are steadily rising, more dietary investigations are necessary. The use of validated FFQs in conjunction with other instruments like biomarkers, 24-hour recalls and/or food records is likely to provide more accurate dietary estimations.

In conclusion, it is essential that researchers develop well-designed, validated FFQs that are adapted for the GCC to standardise dietary assessments across studies.

### Funding
The authors received no funding for this work.

### Competing Interests
The authors declare there are no competing interests.

### Author Contributions
- Rukshana Hoque conceived and designed the experiments, performed the experiments, analyzed the data, prepared figures and/or tables, authored or reviewed drafts of the paper, and approved the final draft.
- Erin Strotheide performed the experiments, analyzed the data, prepared figures and/or tables, authored or reviewed drafts of the paper, and approved the final draft.
- Juliann Saquib conceived and designed the experiments, authored or reviewed drafts of the paper, and approved the final draft.
- Nazmus Saquib conceived and designed the experiments, authored or reviewed drafts of the paper, supervised the study, and approved the final draft.

### Data Availability
As a literature review, there is no data set associated with this article.

## Supplemental Information

Supplemental information for this article can be found online at http://dx.doi.org/10.7717/peerj.10163#supplemental-information.

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
