# Peer review of "Assessment of nationally representative dietary studies in the Gulf Cooperation Council: a scoping review"

_PeerJ, doi:10.7717/peerj.10163_

## Round 0.1 · original submission · Major Revisions

Dear Dr Saquib

Reviewers have commented on your literature review manuscript. They have found some merits in the work, however, they identified some key areas that do require your attention to address carefully.

I believe your work has some merit as a literature review. However, kindly consider providing further details on the synthesis of contents/contexts of key cited literature. It is not only to cite literature, it is about the synthesis, the hows? the whys? Consider these in addition to the reviewers' comments.

Looking forward to your feedback, improved revised manuscript. If you need more time, do not hesitate to communicate with the PeerJ.

Thank you for your support to PeerJ.

Charles O.R. Okpala
Academic Editor

Reviewer 1 ·

Basic reporting

The manuscript describes a review to collect, quantify and critique the quality of data on the dietary studies conducted in GCC countries by using an objective scoring system approach.
It was well written and easy to follow. However, the authors could avoid the use of 'we' in the write up and rather use reporters' speech.

Experimental design

The study design/methodology is clear. However, it would be useful to justify why only two databases (PubMed and Web of Science) were used for the search.

Validity of the findings

The authors indicated the limitations of this review due to the fact that only two main databases were used. Thus the limited number of studies might have affected the findings.

Additional comments

The review is interesting and easy to follow. The recommendations are sound and would be useful. However, it would be useful to extend the review including other search database such as DOAJ, GOOGLE SCHOLAR and SCOPUS.

Reviewer 2 ·

Basic reporting

The article is well written for the most part. I made some recommendation on some words that should be replaced and some sentences need to be rephrased or separated to ensure good flow. See some examples below:

1. Line 2 and 3: I recommend writing GCC in full in the title
2. Line 39-42: should be divided into two complete sentences. Replace “i.e” with “that is”
3. Line 48: replace “e.g” with “such as”
4. Line 53-54: please rephrase this sentence “However, dietary studies have been limited; the research output from Arab countries constitutes ≈ 1% of global research (9).”
5. Line 171: replace “sugared” with either “sweetened” or “sugary”

The authors have provided relevant literature references, standard article structure and the introduction clearly introduced the subject and the motivation behind it.
However, I think the introduction would be more valuable to readers if more information are provided on the subject of obesity in the GCC, and some of the factors limiting research quantity and quality on this subject.
It seems also that the authors did not point out any research similar to what they have conducted. This makes it difficult to fully assess the quality of this study.

Experimental design

It is a bit unclear if this article is within the Aims and scope of the journal. Although, the manuscript has been considered as a literature review, the structure is closer to a research article (because it has methods, results and discussion which are mostly absent in a typical review manuscript). The manuscript also looks like a meta-analysis. I'll leave it to the editor to determine if this is actually a research and review article and my feedback will based on evaluation of the manuscript as a literature review.
I don't think the manuscript is detailed enough to be considered a review of the literature. Most of the cited literature were not explored in detail. To make it better, the authors could include all the published articles they found in their search into this review (regardless of their strength and weaknesses) and pointing out those studies could be improved. That, I think would a better read.

Validity of the findings

The findings of the search of the literature conducted by the authors were clear and the conclusions identified gaps and future direction.

Additional comments

No comment

---

## Round 0.2 · accepted · Accept

The reviewers are now satisfied with the improvements authors have made. All queries raised have been adequately addressed. The work can now be accepted for publication. Thank you for your scholarly contribution to PeerJ. We look forward to your future submissions.

Reviewer 1 ·

Basic reporting

The authors have managed to address the review comments and improved the manuscript.

Experimental design

Satisfactory!

Validity of the findings

No comment.

Reviewer 2 ·

Basic reporting

Improvements recommended during the first review has been applied

Experimental design

Study design has been improved with the addition of the supplemental files

Validity of the findings

Conclusions and next steps were clearly stated

Additional comments

No comment